# Frequent 4EBP1 Amplification Induces Synthetic Dependence on FGFR Signaling in Cancer

**DOI:** 10.3390/cancers14102397

**Published:** 2022-05-13

**Authors:** Prathibha Mohan, Joyce Pasion, Giovanni Ciriello, Nathalie Lailler, Elisa de Stanchina, Agnes Viale, Anke van den Berg, Arjan Diepstra, Hans-Guido Wendel, Viraj R. Sanghvi, Kamini Singh

**Affiliations:** 1Cancer Biology and Genetics Program, Memorial Sloan Kettering Cancer Center, New York, NY 10065, USA; prathibhamohanp@gmail.com (P.M.); pasionj@mskcc.org (J.P.); wendelh@mskcc.org (H.-G.W.); 2Department of Computational Biology, University of Lausanne, CH-1005 Lausanne, Switzerland; giovanni.ciriello@unil.ch; 3Integrated Genomics Operation, Marie-Josée and Henry R. Kravis Center for Molecular Oncology, Memorial Sloan Kettering Cancer Center, New York, NY 10065, USA; contact@nathalielailler.com (N.L.); vialea@mskcc.org (A.V.); 4Molecular Pharmacology Program, Memorial Sloan-Kettering Cancer Center, New York, NY 10065, USA; destance@mskcc.org; 5Department of Pathology and Medical Biology, University of Groningen, University Medical Center Groningen, 9713 GZ Groningen, The Netherlands; a.van.den.berg01@umcg.nl (A.v.d.B.); a.diepstra@umcg.nl (A.D.); 6Department of Molecular and Cellular Pharmacology, Sylvester Comprehensive Cancer Center, Miller School of Medicine, University of Miami, Miami, FL 33136, USA; vxs542@med.miami.edu; 7Department of Molecular Pharmacology, Albert Einstein College of Medicine, Albert Einstein Cancer Center, Bronx, NY 10461, USA

**Keywords:** eIF4E, 4EBP1, FGFR1, breast cancer, lung cancer, ribosome footprinting

## Abstract

**Simple Summary:**

Our work establishes that amplification of 4EBP1, as a part of Chr. 8p11, creates a synthetic dependency on FGFR1 signaling in cancer. 4EBP1 is phosphorylated by FGFR1 and PI3K signaling, and accordingly cancer with 4EBP1-FGFR1 amplification is more sensitive to FGFR1 and PI3K inhibition due to inhibition of 4EBP1 phosphorylation. Moreover, we characterize the translational targets of 4EBP1 and identify that 4EBP1 specifically regulates the translation of genes involved in insulin signaling, glucose metabolism, and the inositol pathway that plays a role in cancer progression.

**Abstract:**

The eIF4E translation initiation factor has oncogenic properties and concordantly, the inhibitory eIF4E-binding protein (4EBP1) is considered a tumor suppressor. The exact molecular effects of 4EBP1 activation in cancer are still unknown. Surprisingly, 4EBP1 is a target of genomic copy number gains (Chr. 8p11) in breast and lung cancer. We noticed that 4EBP1 gains are genetically linked to gains in neighboring genes, including WHSC1L1 and FGFR1. Our results show that FGFR1 gains act to attenuate the function of 4EBP1 via PI3K-mediated phosphorylation at Thr37/46, Ser65, and Thr70 sites. This implies that not 4EBP1 but instead FGFR1 is the genetic target of Chr. 8p11 gains in breast and lung cancer. Accordingly, these tumors show increased sensitivity to FGFR1 and PI3K inhibition, and this is a therapeutic vulnerability through restoring the tumor-suppressive function of 4EBP1. Ribosome profiling reveals genes involved in insulin signaling, glucose metabolism, and the inositol pathway to be the relevant translational targets of 4EBP1. These mRNAs are among the top 200 translation targets and are highly enriched for structure and sequence motifs in their 5′UTR, which depends on the 4EBP1-EIF4E activity. In summary, we identified the translational targets of 4EBP1-EIF4E that facilitate the tumor suppressor function of 4EBP1 in cancer.

## 1. Introduction

Genomics of human breast and lung cancer revealed several chromosomal aberrations and mutations, including amplification of Chr. 8p11-12 in about 15% of breast cancer and 18% of lung cancer cases [1,2,3]. Chr. 8p11-12 includes a major oncogene called FGFR1 and more strikingly 4E-BP1 [1,2,3]. FGFR1 promotes tumor growth and proliferation by activating growth and nutrient signaling via PI3K and mTOR activation [4,5,6].

The 4E-binding protein (4E-BP1) is a tumor suppressor and blocks the oncogenic eIF4E initiation factor through its direct interaction and competition for eIF4E binding to the eIF4F complex [7,8]. During mTOR/AKT/PI3K activation, 4E-BP1 is phosphorylated, which frees and activates eIF4E and stimulates translation and tumor growth in vivo [9]. Free eIF4E acts as a proto-oncogene, and its overexpression led to lymphoma formation in mouse models [8]. Direct evidence for 4E-BP1′s tumor suppressor function comes from *4E-BP1**^−/−^/4ebp2^−/−^* knockout mice that showed increased tumorigenesis in a tobacco-induced lung cancer model [7]. Loss of 4E-BP1 and 4E-BP2 increases tumorigenesis in TP53 knock-out mice [10]. Targeting cap-dependent translation through dominant active mutant of 4E-BP1 that cannot be phosphorylated shows increased sensitivity towards Gemcitabine [11]. EIF4E activation is implicated in cancer growth, and the knockdown of eIF4E reduces breast cancer growth [8,12]. Conversely, an activated eIF4F initiation complex is essential for tumor initiation and maintenance of malignant human breast cancer [13]. Inhibitors targeting upstream PKC and PI3K signaling to induce apoptosis in cancer cells through reducing phosphorylation of 4E-BP1 leads to inhibition of eIF4E and eIF4F activity [14,15]. Insufficient inhibition of 4E-BP1 phosphorylation has been implicated in primary resistance against ATP-competitive mTORC1 inhibitors, suggesting that inactivated 4E-BP1 supports tumor growth [16]. All these studies suggest that 4E-BP1 acts as a tumor suppressor by blocking oncogenic eIF4E and eIF4F activity. Surprisingly, Chr. 8p11-12 amplification co-amplifies 4EBP1 along with FGFR1, suggesting that FGFR1-dependent phosphorylation of 4EBP1 [17] might create a vulnerability to FGFR1 inhibitors in these tumors.

mTOR regulates the translation of ribosomal proteins and promotes cancer initiation and metastases while MYC activation specifically regulates the translation of the mitochondrial respiration complex to support aggressive tumor growth [18,19,20]. EIF4A regulates the translation of the RNA G-quadruplex containing mRNAs including MYC and KRAS oncogenes [21,22]. EIF4E regulates the translation of genes related to ROS and induces transformation and survival in cancer cells [23]. EIF4E also regulates the translation of lipid metabolism in a high-fat diet-induced model of obesity [24]. Many studies characterize the effect of mTOR, oncogenes, EIF4A, and EIF4E on translation; however, the specific effects arising from modulating 4EBP1 on mRNA translation in cancer are not well-defined. In this study, we explored the contribution of FGFR1/4EBP1 amplification in mediating translation activation and cancer progression in breast and lung cancer. We further explored the therapeutic vulnerability generated by 4EBP1 amplification in breast and lung cancer. Since 4EBP1 activity controls the mRNA translation, using ribosome footprinting analysis, we characterized the genome-wide translational targets of 4EBP1 that may play role in mediating cancer signaling and growth in 4EBP1-amplified cancer.

## 2. Materials and Methods

### 2.1. Cell Culture and Treatment

All the breast and lung cancer cell lines used in this study were obtained from American Type Culture Collection and cultured as per the instructions. ZR75-30, NCIH838, and NCIH1703 cells were cultured in RPMI-1640 cell culture medium supplemented with 10% fetal bovine serum and penicillin-streptomycin. JIMT1 and 293T cells were cultured in DMEM supplemented with 10% fetal bovine serum and penicillin-streptomycin (100 U/mL: 100 μg/mL). CAMA1 cells were cultured in Eagle’s Minimum Essential Medium, (ATCC cat. no. 30-2003) supplemented with 10% fetal bovine serum and penicillin-streptomycin. T47D cells were cultured in RPMI-1640 supplemented with 0.2 Units/mL of bovine insulin (Life Technologies, cat. no. 12585-014), 10% fetal bovine serum, and penicillin-streptomycin (100 U/mL; 100 μg/mL). Fetal bovine serum and penicillin-streptomycin were purchased from Life Technologies Cells and were treated with indicated drugs for indicated time points in the complete media.

### 2.2. Drugs and Inhibitors

FGFR inhibitor BGJ398 was purchased from Selleck Chem (cat. no. S2183). Pan PI3K inhibitors BKM120 (cat. no. S2247) and BLY719 (cat. no. S2814) were purchased from Selleck Chem. Cycloheximide (cat. no. C4859), and doxycycline (cat. no. D3447) were purchased from Sigma.

### 2.3. Global mRNA Translation

Cultured cells were labeled with Cy5-conjugated puromycin (5 uM) for one hour at the end of the indicated time point following drug treatment. Cy5-conjugated puromycin is readily incorporated into live cells without the need for methionine starvation. Changes in mean fluorescence intensity of Cy5-conjugated puromycin as a measure of newly produced protein were analyzed by flow cytometric analysis. For nascent protein labeling in 293T cells, we used Click-iTR AHA (L-azidohomoalanine) metabolic labeling reagent purchased from Invitrogen (cat no. C10102) and following the manufacturer’s instructions. Briefly, cells were incubated in a methionine-free medium for 30 min before AHA labeling for 1 h after doxycycline treatment (2 mg/mL). Cells were fixed with 4% paraformaldehyde (wt/vol) in PBS for 15 min and permeabilized with 0.25% Triton X-100 (vol/vol) in PBS for 15 min followed by one wash with 3% BSA. Cells were then stained using Alexa Fluor 488 Alkyne (Invitrogen cat no. A10267) using Click-iT Cell reaction Buffer Kit (Invitrogen cat no. C10269). Changes in mean fluorescence intensity (MFI) of Alexa Fluor 488 Alkyne staining were detected by flow cytometric analysis and used as a measure of newly synthesized protein.

### 2.4. Generation of 4EBP1 CRISPR-cas9-Expressing Cells

CRISPR-targeting Exon 1 of 4EBP1 was purchased from Sigma as one plasmid system in u6-sgRNA-pCMV-cas9-RFP. JIMT1 and A549 cells were transiently transfected with 4EBP1 CRISPR-cas9-RFP plasmid. RFP-positive cells were sorted after 48 h of transfection. 4EBP1 knockdown was assayed by nuclease surveyor assay using the kit and instructions provided in the kit (Surveyor^®^, Tokyo, Japan, Mutation Detection Kit—S100 cat. no. 706020 from Integrated DNA Technologies). Knockdown in 4EBP1 protein was confirmed by Western blotting analysis for a total of 4EBP1.

### 2.5. Generation of Doxycycline-Induced 4EBP1-4A Plasmid

A gene-block encoding human 4EBP1 variant with alanine substitutions at T37, T46, S65, and T70 was cloned in LT3REVIR lentiviral backbone between BamH1 and EcoRI sites [25]. For lentiviral-mediated transduction, we used 293T cells to generate the virus and concentrated the virus using a Lenti-X concentrator (Takara cat. no. 631232). Successfully transduced cells were sorted based on GFP expression and used for downstream experiments.

### 2.6. Ribosome Footprinting

Human 293T cells expressing Dox-induced 4EBP1-4A plasmid were treated with DMSO or doxycycline (2 mg/mL; 12 h) followed by cycloheximide treatment for 10 min. Total RNA and ribosome-protected fragments were isolated following a published protocol [26]. Deep sequencing libraries were generated from these fragments and sequenced on the HiSeq 2000 platform. Genome annotation was from the human genome sequence GRCh37 downloaded from Ensembl public database: http://www.ensembl.org accessed on 10 April 2021.

### 2.7. Sequence Alignment

Sequence alignment for total RNA and ribosome footprinting reads was carried out as described in our previous published study [21]. Briefly, ribosome footprint (RF) reads were filtered based on the quality score and trimmed for the linker sequence (5′-CTGTAGGCACCATCAAT-3′). To remove ribosomal RNA, the footprint reads were then aligned to the ribosome RNA sequences of GRCh38 downloaded from UCSC Table Browser (https://genome.ucsc.edu/cgi-bin/hgTables, accessed on 10 April 2021). After removing the reads aligned to the ribosome RNAs, RF reads were mapped to the human genome sequence GRCh38 downloaded from Ensembl public database, http://www.ensembl.org (accessed on 10 April 2021), using RNA Star with default parameters. Total mRNA sequencing reads were aligned to the GRCh38 reference using RNA Star. Only uniquely aligned reads and reads aligned to the exonic regions of the protein-coding genes were used for further analysis.

### 2.8. Footprint Profile Analysis Using Ribo-Diff

We used Ribo-diff to analyze the translation efficiency based on the ribosome footprinting and mRNA sequencing data. Genes with significantly changed translation efficiency were defined by the *q*-value cut-off equal to 0.05.

### 2.9. RNA Motif Analysis

For RNA motif analysis, we used the 5′UTR sequence of the longest transcript selected for each gene as described in our previous study [21,22]. Both the significant genes with increased or decreased TE and the corresponding background gene sets were used to predict motifs by DREME [27]. The occurrences of the significant motifs (E < 0.05 and *p* < 1 × 10^−8^ from DREME) were called using FIMO [27] with default parameters for strand-specific prediction of all the 5′UTR sequences.

### 2.10. Immunoblots

Lysates were made using RIPA lysis buffer (1X RIPA lysis buffer consists of 50 mM Tris HCl, 150 mM NaCl, 1.0% (*v*/*v*) NP-40, 0.5% (*w*/*v*) sodium deoxycholate, 1.0 mM EDTA, 0.1% (*w*/*v*) SDS, and 0.01% (*w*/*v*) sodium azide at a pH of 7.4) with 0.5M EDTA and proteases and phosphatases. Sixty micrograms of protein were loaded onto SDS-PAGE gels and then transferred onto Immobilon-FL Transfer Membranes (Millipore cat. no. IPFL00010). The antibodies used were p-4EBP1 (Cell Signaling Technology cat. no. 2855L), 4EBP1 (Cell signaling Technology 9644S), FGFR1 (cat. no. ab10646), HK1 (cat. no 2804S), HK2 (cat. no 2867S), GAPDH (Cell signaling Technology cat. no 5174), and β-actin (Sigma cat. no A5316). Quantification of the protein expression was carried out using ImageJ software. Each protein was normalized to β-actin and fold change was compared to the untreated samples or the FGFR1-4EBP1 wild-type cell line, as indicated in each figure.

### 2.11. Human Breast Cancer Cell Line Xenografts

Human breast cancer JIMT wild-type or JIMT1 4EBP1-CRISPR-cas9-expressing cells were injected into subcutaneous flank in J: Nu mice (5 million cells per flank). Tumors were monitored and measured using a vernier caliper twice a week. When tumors were between 80 and 100 mm^3^ (mean dimension of 0.2–0.3 cm), vehicle or BKM120 (25 mg/kg) was injected in mice intraperitoneally twice a week until the control mice developed fully grown tumors. *p*-values were calculated using 2-way repeated-measures ANOVA. All animal experiments were performed by regulations from Memorial Sloan Kettering Cancer Center’s Institutional Animal Care and Use Committee.

### 2.12. Clonogenic Survival Assay

To evaluate the long-term survival potential of the cells, we performed a clonogenic survival assay for respective cells. Briefly, 20 × 10^3^ cells were seeded per well in 6-well plates and allowed to adhere overnight in regular growth media. Vehicle (DMSO) or BKM120 (1 μM) or BGJ398 (50 nM) was added and refreshed every 3 days until the end of the experiment (14 days). At the end of the treatment, cells were fixed in 4% formaldehyde (*wt*/*vol*) for 15 min at RT and subsequently stained with 0.1% crystal violet (*wt*/*vol*) and digitalized on an image scanner. All experiments were performed at least three times in triplicate, and representative results are shown.

### 2.13. Real-Time PCR Assay

Total RNA was extracted using AllPrep DNA/RNA/Protein Mini Kit (Qiagen 80004). cDNAs were synthesized from 1 μg of total RNA using SuperScript III First-Strand (Invitrogen 18080-400) and were amplified using Taqman Universal Master Mix II, no UNG (Applied Biosystems 4427788). Analysis was performed by ΔΔCt. Applied Biosystems Taqman Gene Expression Assays: human HK1 Hs00175976_m1, HK2 Hs00606086_m1, and Beta-Actin 4332645. Relative mRNA expression was evaluated after normalization for beta-actin expression. Data show results from at least two independent experiments.

### 2.14. Luciferase Assay

We generated a luciferase reporter assay by cloning the identified motifs into the 5′UTR of Renilla luciferase plasmid pGL4.73. bGal reporter plasmids were used as internal controls. Luciferase assays were performed using the Dual-Luciferase Reporter Assay System (Promega E1960) following the manufacturer’s instructions.

Motifs sequence:

Motif-1 (GC-rich):

GGCGTCGGCG-GCGGCGGCGGCAGCGGCTCCG-GCCGAGGTGC.

Control Motif:

TTGTTGGTTT-TCTTTTTCTTT-AAATTAAAAA-ATAAAAGAAAA.

### 2.15. Statistical Analysis

All the results were analyzed with two-tailed t-tests unless specified. The significance of motif enrichments was from the DREME program based on Fisher’s exact test. A hypergeometric test was performed to test for the significance of the enrichment of the gene overlap in the KEGG pathway.

Online Content: Supplementary display items are available in the online version of the paper. Raw and processed data for the ribosome footprinting and total mRNA sequencing were deposited in the NCBI Gene Expression Omnibus database at Pubmed (GSE accession number GSE197735).

## 3. Results

### 3.1. 4EBP1 Is a Target of Chr. 8p11 Amplification in Breast and Lung Cancer

EIF4EBP1 (4EBP1) is located at Chr. 8p11. Genomics analysis for copy number revealed that Chr. 8p11 is frequently amplified in breast cancer with a focal region co-amplifying FGFR1, 4EBP1, and WHSC1L1 (Figure 1A). The copy number of 4EBP1 and FGFR1 showed a strong correlation in breast cancer (Figure 1B). Copy number gain was observed in 15% of the breast cancer samples and was present regardless of subtypes and ER/PR/Her2 status (Figure 1C). Co-amplification of 4EBP1 and FGFR1 was significantly correlated with poor survival in breast cancer with a median survival of 94 months in 4EBP1 and FGFR1 amplified versus 115 months in non-amplified cases (Figure 1D). Next, copy number gains were positively correlated with increased RNA expression (Figure 1E,F and Appendix A). Protein expression of 4EBP1 was increased in the Chr. 8p11-amplified compared to non-amplified cases in both breast and lung cancer patients (Figure 1G and Appendix A). PI3K and PTEN are other frequently mutated targets in breast and lung cancer [28,29]. We observed that PI3K and PTEN mutations were mutually exclusive with FGFR1-4EBP1 amplification in both breast and lung cancer, suggesting their redundant requirement (Figure 1H and Appendix A). Next, we observed that 4EBP1 was highly phosphorylated in 4EBP1- and FGFR1-amplified versus non-amplified cases at all the three phosphorylation sites (Thr37, Thr70, and Ser65) (Figure 1I and Appendix A). These observations suggest that FGFR1 gains act to attenuate the function of 4EBP1 via PI3K-mediated phosphorylation at Thr37/46, Ser65, and Thr70 sites.

### 3.2. 4EBP1-Amplified Tumors Show Increased Sensitivity to FGFR1 and PI3K Inhibition

Human cancer cell lines showed co-amplification of FGFR1-4EBP1 (6–9%), and this was mutually exclusive with PI3K mutations observed in 20% of the samples in Cell Line Encyclopedia (Appendix A). We picked a few breast and lung cancer cell lines and validated the amplified status of 4EBP1 and FGFR1 proteins by Western blotting analysis (Appendix A and Figure 2A,B). Next, we showed that FGFR1-4EBP1-amplified cells JIMT1 and NCIH1703 showed increased sensitivity to FGFR1 inhibitor BGJ398 and pan-PI3K inhibitor BKM120 when compared to FGFR1-4EBP1 non-amplified cells (Figure 2C,D). Database from DepMAP also revealed that FGFR1-4EBP1-amplified cells were more sensitive to the FGFR1 inhibitor ponatinib compared to FGFR1-4EBP1-non-amplified cells in both breast and lung cancer cell lines (Figure 2E,F). PI3KCA signaling was downstream of FGFR1, and therefore PI3KCA-mutated cells showed reduced sensitivity to the FGFR1 inhibitor ponatinib when compared to wild-type PI3KCA cells in both breast and lung cancer cell lines, as observed in the database from DepMAP (Figure 2G,H). These data suggest that 4EBP1-FGFR1-amplified tumors show increased sensitivity to FGFR1 and PI3K inhibition.

### 3.3. Loss of 4EBP1 Results in Activated Translation and Reduced Sensitivity to FGFR1 and PI3K Inhibition

Next, we measured the global translation levels using OP-puromycin-Cy5 labeling in breast cancer cells treated with FGFR1 and PI3K inhibitors. FGFR1-4EBP1-non-amplified cells ZR7530 showed no reduction in global translation while FGFR1-4EBP1-amplified cells JIMT1 showed a significant reduction in global translation following FGFR1 and PI3K inhibitor treatment for 2 hr (Figure 3A). The Western blotting analysis further confirmed reduced p-4EBP1 levels in 4EBP1-amplified JIMT1 cells following FGFR1 and PI3K inhibitor treatment for 24 hr (Figure 3B,C). To further characterize the effect of 4EBP1, we generated CRISPR-Cas9-mediated knockdown of 4EBP1 in JIMT1 and A549 cells. In a mixed population of 4EBP1-CRISPR-Cas9-induced cells, we confirmed the gene editing and knockdown of 4EBP1 by nuclease surveyor and Western blotting assay (Appendix A). We used these selected populations of 4EBP1-CRISPR-Cas9-edited cells for further experiments. We observed that 4EBP1-CRISPR-Cas9-edited JIMT1 cells showed increased global translation that was not inhibited by FGFR1 and PI3K inhibitors compared to 4EBP1-amplified parental JIMT1 cells (Figure 3D). Further, 4EBP1-CRISPR-Cas9-edited JIMT1 cells showed decreased sensitivity to FGFR1 and PI3K inhibitors compared to 4EBP1-amplified parental JIMT1 cells as measured by a short-term viability assay and long-term colony survival assay (Figure 3E,F). Next, we implanted JIMT1 parental and 4EBP1-CRISPR-Cas9-edited cells subcutaneously in nude mice and observed increased tumor growth and weight in 4EBP1-CRISPR-Cas9-edited JIMT1 cells (Figure 3G–I). Tumors generated from 4EBP1-CRISPR-Cas9-edited JIMT1 cells showed significantly reduced sensitivity to the PI3K inhibitor BKM120 (Figure 3J,K). These data suggest that 4EBP1 in the FGFR1-amplified setting can feed tumor growth by activating mRNA translation and increasing the sensitivity to FGFR and PI3K inhibition (Figure 3L).

### 3.4. Ribosome Footprinting Identifies Translational Targets of 4EBP1

To identify the translational targets of 4EBP1, we generated 293T cells stably expressing a doxycycline-inducible phospho-mutant 4EBP1 that cannot be phosphorylated by mTOR and PI3K, referred to as 4EBP1-4A (Thr37/40, Thr70, and Ser65 are all mutated to alanine). Phospho-4EBP1 expression was reduced following doxycycline treatment at 12, 24, and 48 h while the total 4EPB1 levels were induced due to overexpression of 4EBP1-4A (Figure 4A). The global translation was measured by AHA labeling and showed a 50% reduction at 24 h following doxycycline treatment (Figure 4B). mRNA expression analysis showed induced 4EBP1-4A at 12 h following doxycycline treatment (Appendix A). We observed a reduction in the proliferation rates of 293T cells expressing 4EBP1-4A following doxycycline treatment (Appendix A). To comprehensively identify mRNAs that depend on eIF4A for translation, we performed ribosome footprinting and deep sequencing in the presence and absence of 4EBP1-4A following 24 h of the doxycycline treatment. Briefly, we normalized ribosome-protected RNA fragments (RF reads) to the total RNA abundance to isolate changes in translation efficiency (TE). We performed ribosome footprinting on three control (DMSO) and three doxycycline (24 h)-treated 293T samples (Figure 4C).

Read mapping and sample correlation were carried out as described in our previous study [21,22]. The replicates showed significant correlations among the replicates with Pearson coefficients of >0.96 and >0.6 in RNA-seq and Ribo-seq samples, respectively (Appendix A).

We used the RiboDiff statistical framework to isolate the effect on mRNA translation ^2^. With a very stringent statistical cut-off at *q* < 0.05 (FDR < 5%), we identified 197 mRNAs whose translation was significantly repressed (TE-down: *n* = 197; *q* < 0.05), and we also detected a set of mRNAs showing a relative increase in ribosome occupancy (TE-up: *n* = 92; *q* < 0.05) (Figure 4D) (Complete dataset is available at GSE197735.). A full list of genes differentially affected by TE in 4EBP1-4A-expressing 293T cells is provided in Appendix A. We noticed that 4EBP1-dependent mRNAs showed a significant reduction (TE-down) or increase (TE-up) in the ribosome coverage on the mRNA (Figure 4E and Appendix A). Unaffected mRNAs referred to as background did not show any changes in the ribosome coverage (Appendix A).

### 3.5. 4EBP1 Controls the Translation of Genes Involved in Insulin Signaling, Glucose Metabolism, and Inositol Pathway

Next, we performed a GSEA analysis of the TE-down genes (*n* = 197; *q* value < 0.05) and TE-up genes (*n* = 92; *q* value < 0.05). TE-down genes showed KEGG pathway enrichment for insulin signaling, GNRH signaling, and ribosome KEGG pathways (Figure 5A). TE-up genes showed KEGG pathway enrichment for insulin signaling, ribosome, glycolysis and gluconeogenesis, phosphatidylinositol signaling, and inositol phosphate metabolism (Appendix A). Specifically, 4EBP1-4A repressed the translation of HK2, PRKAA2 (AMPK2), and PRKAR2A (PKR2) while upregulating the translation of HK1, PRKACA, PCK1, and PPP1CC (Figure 5B). 4EBP1-4A also upregulated the translation of IMPA1, PLCG1, ITPK1, and IPPK, which are involved in inositol phosphate metabolism (Appendix A). Western blot shows downregulation of HK2 and upregulation of HK1 protein (Figure 5C). Real-time RT-PCR showed a slight reduction in mRNA expression of HK1 and HK2 following doxycycline treatment while 4EBP1-4A showed increased expression, as expected (Appendix A). Next, we explored the significant TE-down mRNAs for common molecular features. We performed a de novo motif search using the MEME suite. We compared the TE-down group of mRNAs with annotated 5′UTRs (*n* = 350; *p* < 0.05) and a background list of 639 equally expressed and annotated mRNAs that showed no significant change in their translation. We identify 10 sequence motifs that were significantly enriched (*p* < 0.05) in the 5′UTR of TE-down mRNAs when compared to the background (Bkg) mRNAs (Figure 5D). Next, we compared the translation activity of control and GC-rich motif-driven translation using a luciferase reporter assay in 4EBP1-4A-expressing 293T cells. We observed that compared to the control motif, 4EBP1-4A repressed the translation of the GC-rich motif (Figure 5E). In the summary, we identify the translational targets of 4EBP1 and the GC-rich RNA sequence motifs in the 5′UTR that may regulate the translation of 4EBP1-dependent mRNAs, including insulin signaling proteins (Figure 5F).

## 4. Discussion

Tumor suppressor genes are not usually considered therapeutic targets, and once they are mutated, it becomes impossible to reactivate them. This applies to classical tumor suppressors defined by their genomic inactivation (e.g., the two-hit hypothesis) [30]. However, other tumor suppressor genes are defined through their function. For instance, loss of 4E-BP1 promotes cancer development in different animal models [10,11]. These non-classical tumor suppressors are often functionally inactivated by increased degradation or phosphorylation in human cancers. This provides a potential opportunity for their reactivation by inhibiting relevant kinases. We showed that this is the case here—a tumor suppressor was included in a genomic amplified region. The Chr. 8p11-12 amplicons included the oncogenic FGF receptor and the tumor-suppressive 4E-BP1 genes. This co-amplification indicates a direct relationship between FGFR and 4E-BP1, such that FGFR can silence 4EBP1, in this case through phosphorylation, and therefore can be reactivated by using FGFR1 or PI3K inhibitors. Accordingly, we showed that Chr. 8p11-12-amplified breast and lung cancer were highly sensitive to FGFR1 and PI3K inhibitors, and this effect was abolished in 4EBP1 CRISPR-cas9 knockout cells.

Next, through ribosome footprinting analysis, we characterized the specific effect of 4EBP1 on mRNA translation. Inactivating 4EBP1-EIF4E resulted in the downregulation of genes involved in insulin signaling, such as HK2, while upregulating HK1. HK2 is overexpressed in cancer and mediates aerobic glycolysis, tumor growth, and metastases [31,32]. Genetic ablation of HK2 results in tumor growth inhibition in mouse models [33]. Decreased expression of HK1 accelerates tumor malignancy through regulating energy metabolism, and there is an inverse relationship in the expression of HK1 and HK2 in cancer [34]. Differential effects of 4EBP1-4A on the translation of HK1 and HK2 suggest that 4EBP1 may regulate glucose and insulin metabolism to inhibit tumor growth, and this needs to be explored. On the other hand, inactivating 4EBP1-EIF4E resulted in increased translation of genes involved in inositol phosphate metabolism and the phosphatidylinositol signaling system that is implicated in insulin signaling and cell migration in cancer cells [35,36]. Overall, our data indicate that 4EBP1 controls the translation of genes involved in insulin signaling, glucose metabolism, and the inositol pathway. Increased 4EBP1 expression protects from diet-induced obesity and insulin resistance in mouse models [37]. The phenotypic effect of 4EBP1 through translational regulation of these pathways remains to be explored. Our work establishes that inhibiting FGFR1 and PI3K signaling in 4EBP1-amplified tumors could provide a therapeutic advantage, potentially through abrogating 4EBP1-dependent mRNA translation.

## 5. Conclusions

Our study concludes that the gain of Chr. 8p11 co-amplifies FGFR1 and 4EBP1 genes in breast and lung cancer. FGFR1-4EBP1 amplification was mutually exclusive to PI3K mutation. Both FGFR1 and PI3K signaling phosphorylated 4EBP1, resulting in activation of EIF4E and mRNA translation. Chr. 8p11 amplification created a synthetic dependency on FGFR1 and PI3K inhibition due to the regulation of 4EBP1-dependent mRNA translation. Further, we showed that loss of 4EBP1 enhanced tumor growth of FGFR1-4EBP1-amplified cancer cells and reduced their sensitivity to FGFR1 and PI3K inhibition. 4EBP1 specifically regulated the translation efficiency of a subset of mRNAs that includes protein production of key enzymes involved in glucose metabolism, insulin signaling, and the inositol pathway. In summary, our study establishes the translational targets of 4EBP1 and the therapeutic advantage of FGFR1 and PI3K inhibitors in 4EBP1-amplified breast and lung cancer.

## Figures and Tables

**Figure 1 cancers-14-02397-f001:**
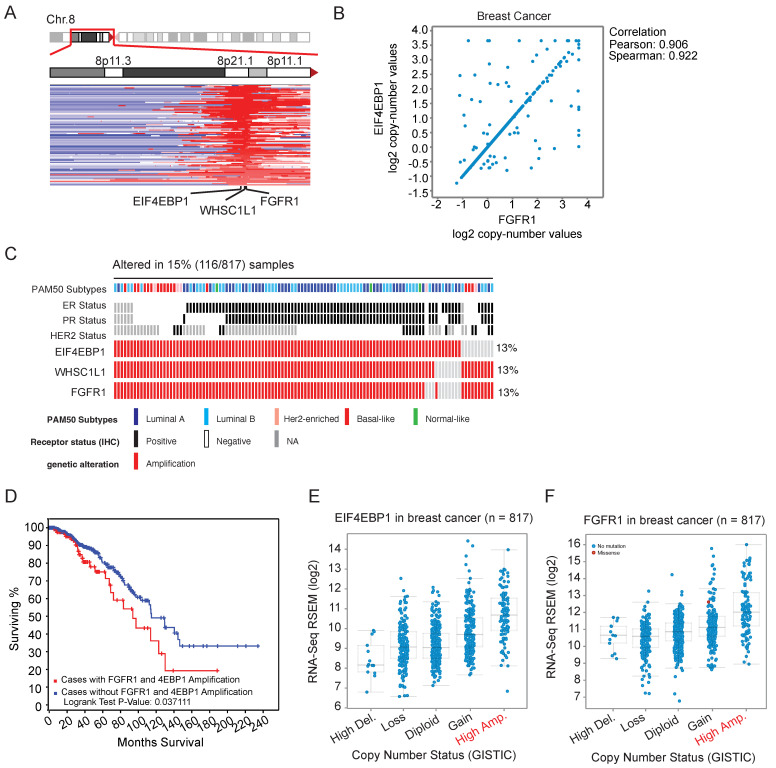
4EBP1 is a target of genomic copy number gains (Chr. 8p11) in breast and lung cancer. (**A**) Visualization of Chr. 8p11-12 region amplified in human breast cancer patients (TCGA data and cbio portal data from MSKCC). Red shows the amplified region while blue is deleted. (**B**) Correlation plots show a positive correlation between 4EBP1 and FGFR1 copy numbers in breast cancer patient samples (TCGA and cBio portal data from MSKCC). (**C**) Oncoprint map showing the frequency of amplification of 4EBP1, WHSC1L1, and FGFR1 in breast cancer TCGA data classified by various subtypes. (**D**) 4EBP1- and FGFR1-amplified breast cancer patients showed poor survival compared to non-amplified cases. (**E**,**F**) Correlation plots show a positive correlation of mRNA expression with copy number of 4EBP1 and FGFR1 in breast cancer patient samples (TCGA and cbio portal data from MSKCC). (**G**) 4EBP1 protein expression was elevated and correlated positively with the mRNA levels in 4EBP1-amplified breast cancer patient samples. (**H**) Oncoprint map showing that 4EBP1 and FGFR1 amplification was mutually exclusive to PI3KCA and PTEN mutation in breast cancer patients. (**I**) Reverse-phase protein array (RPPA) analysis shows that a total 4EBP1, as well as phospho-4EBP1 (T37, T70 and S65) protein, was upregulated in 4EBP1- and FGFR1-amplified breast cancer, patients.

**Figure 2 cancers-14-02397-f002:**
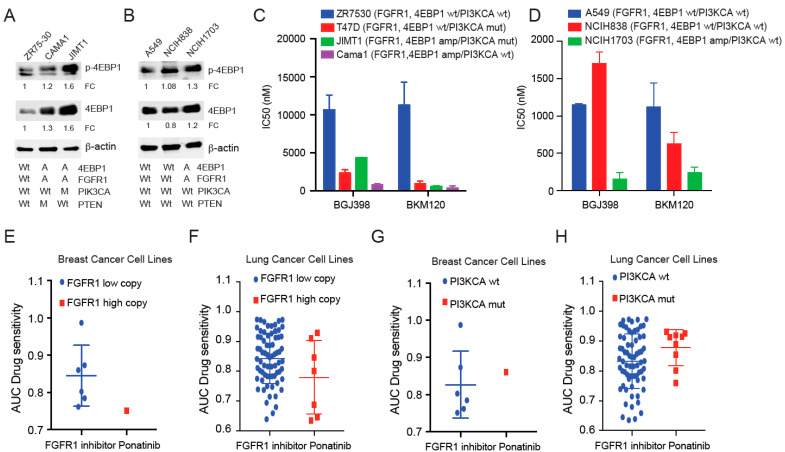
Loss of 4EBP1 results in activated translation and reduced sensitivity to FGFR1 and PI3K inhibition. (**A**,**B**) Immunoblot analysis shows 4EBP1, p-4EBP1 (Ser 65), p-FGFR1, and FGFR1 levels in breast (**A**) and lung cancer (**B**) cell lines with wild-type or amplified 4EBP1. b-actin was used as the loading control. Quantification of respective proteins is indicated as fold change (FC) normalized to b-actin and FGFR1-4EBP1 wild-type cells. The uncropped blots for all the experiments are shown in Appendix A. (**C**,**D**) IC50 analysis for FGFR1 inhibitor BGJ398 and pan-PI3K inhibitor BKM120 in levels in breast (**C**) and lung cancer (**D**) cell lines with wild-type or amplified 4EBP1. (**E–H**). DepMap dataset showing the sensitivity to FGFR1 inhibitor ponatinib in FGFR1-amplified (**E**,**F**) and PI3K-mutant (**G**,**H**) compared to wild-type cells in a panel of breast and lung cancer cell lines.

**Figure 3 cancers-14-02397-f003:**
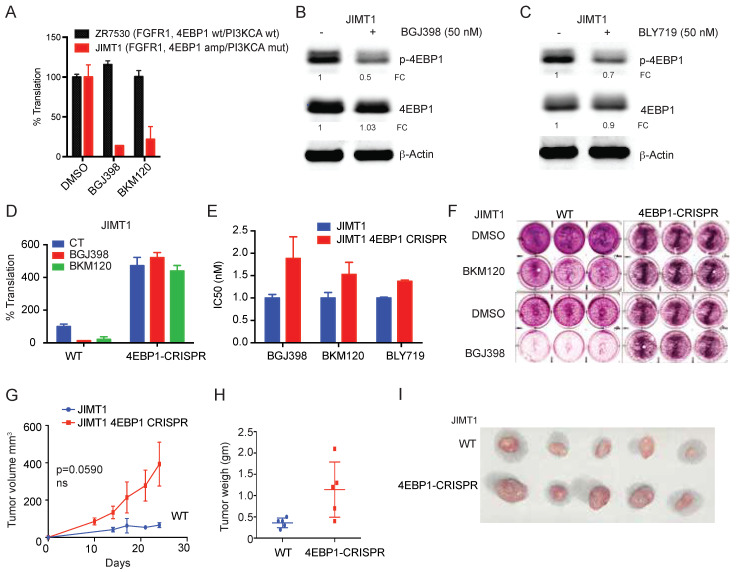
4EBP1 Amplified tumors show increased sensitivity to FGFR1 and PI3K inhibition. (**A**) Global mRNA translation as measured by OP-Puro-Cy5 incorporation in ZR7530 (4EBP1 wild-type) and JIMT1 (4EBP1-amplified) breast cancer cells. (**B**,**C**) Immunoblot showed that BGJ398 (50 nM, 24 h) and BLY719 (50 nM, 24 h) treatment reduced the phospho-4EBP1 (Ser 65) levels in ZR7530 and JIMT1 breast cancer cells. Total 4EBP1 remained unchanged, and b-actin was used as the loading control. Quantification of respective proteins is indicated as fold change (FC) normalized to b-actin and untreated cells. (**D**) Global mRNA translation as measured by OP-Puro-Cy5 incorporation in JIMT1 wild-type (WT) and 4EBP1-CRISPR breast cancer cells. (**E**) Cell viability assay showed that IC50 was increased in JIMT1 4EBP1-CRISPR-deleted cells compared to wild-type cells in response to BGJ398, BKM120, and BLY719. (**F**) Clonogenic assay showed that JIMT1 4EBP1-CRISPR-deleted cells were less sensitive to pan-PI3K inhibitor BKM120 and FGFR1 inhibitor BGJ398 compared to the wild-type cells. (**G–I**) Tumor growth (**G**), tumor weight (**H**), and images (**I**) of JIMT1 wild-type (WT) and 4EBP1-CRISPR-deleted JIMT1 cells in vivo in a xenograft model show that 4EBP1 deletion enhanced the growth of the tumor. (**J–K**) Tumor growth (**J**) and images (**K**) of JIMT1 wild-type (WT) and 4EBP1-CRISPR-deleted JIMT1 cells treated with BKM120 (25 mg/Kg, twice a week) show that 4EBP1-deleted tumors were less sensitive to PI3K inhibition. (**L**) The diagrammatic representation shows that FGFR1 phosphorylates 4EBP1 through mTOR-PI3K-AKT1 signaling, resulting in eIF4E activation and tumor growth. Loss of 4EBP1 expression or phosphorylation resulted in mRNA translation inhibition and reduced tumor growth, suggesting that FGFR1 is the genetic target of Chr. 8p11 gains, and co-amplification of 4EBP1 increases sensitivity to FGFR1 and PI3K inhibition.

**Figure 4 cancers-14-02397-f004:**
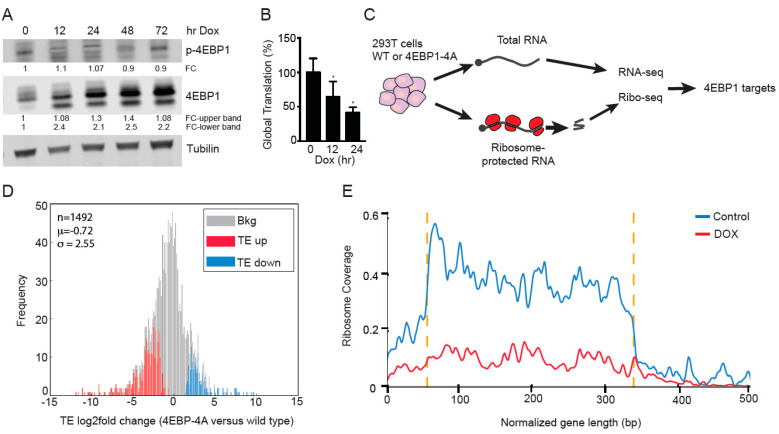
Ribosome footprinting identifies translational targets of 4EBP1. (**A**) Immunoblot showing the overexpression of 4EBP1-4A mutant form in a doxycycline-dependent manner in 293T cells at the indicated time points. Phosphorylation of 4EBP1 at Ser 65 remained reduced in the doxycycline-treated cells compared to the untreated control cells that had basal levels of phosphorylated 4EBP1 (Ser 65). Quantification of respective proteins is indicated as fold change (FC) normalized to b-actin and untreated 293T cells. (**B**) Global mRNA translation was reduced following doxycycline-induced overexpression of 4EBP1-4A mutant protein in 293T cells as measured by AHA incorporation. (**C**) Schematic showing the experimental design for RNA seq and ribosome footprinting in 293T wild-type or 4EBP1-4A-mutant-expressing cells. A comparison of ribosome-protected sequences and total mRNA isolates the translational efficiency for each mRNA (TE). (**D**) Frequency distribution of the change in translation efficiency (TE) in control (WT 4EBP1) and doxycycline-treated (4EBP1-4A mutant) 293T cells. Using the statistical cut-offs of *q* < 0.05, we identified mRNAs with decreased (TE-down, red) and increased (TE-up, blue) and unchanged translation (background, gray) in three biological replicates. (**E**) Ribosome coverage was reduced throughout the mRNA length in the TE-down mRNAs in doxycycline-treated cells compared to the control cells.

**Figure 5 cancers-14-02397-f005:**
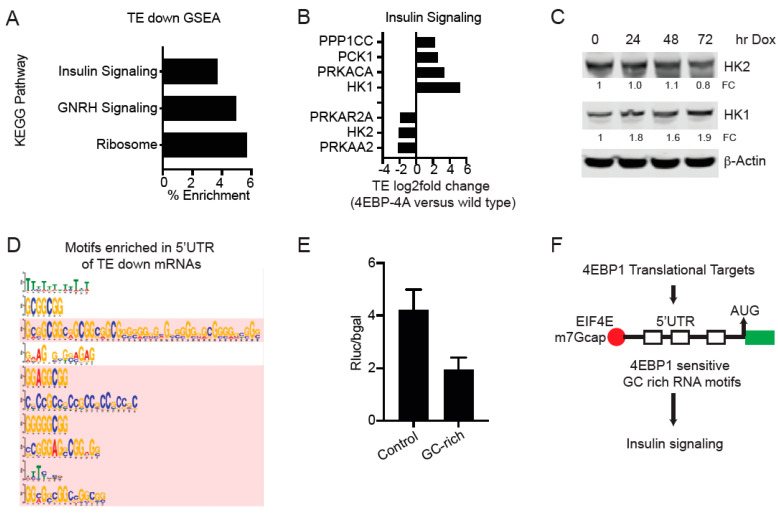
4EBP1 controls the translation of genes involved in insulin signaling, glucose metabolism, and the inositol pathway. (**A**) GSEA KEGG pathway analysis of 4EBP1-dependent (TE-down) genes. (**B**) TE (log2fold change) of key genes involved in insulin signaling. (**C**) Immunoblot validation translational downregulation of HK2 and upregulation of HK1 protein in a 4EBP1-dependent manner. B-actin was used as the loading control. Quantification of respective proteins is indicated as fold change (FC) normalized to b-actin and untreated cells. (**D**). Conserved RNA motifs enriched in the 5′UTR of mRNA that was downregulated in 4EBP1-4A-mutant-expressing cells compared to the background (Bkg) genes. (**E**) Luciferase assay showing relative translation activity of control (AT-rich) and GC-rich motifs in 293T cells expressing 4EBP1-4A (*n* = 6, *p <* 0.05). (**F**) Diagram showing the regulation of 4EBP1-dependent translation through 4EBP1-sensitive RNA motifs in the 5′UTR. 4EBP1 regulated the translation of key proteins involved in insulin signaling.

## Data Availability

Sequencing data generated in this study are available in the NCBI Gene Expression Omnibus database (GSE accession number GSE197735).

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
