# Peer review of "Frequent 4EBP1 Amplification Induces Synthetic Dependence on FGFR Signaling in Cancer"

_cancers, 2022, doi:10.3390/cancers14102397_

Round 1

Reviewer 1 Report

This work investigates FGFR1 gains act to attenuate the function of 4EBP1 via PI3K mediated phosphorylation at Thr37/46, Ser65, and Thr70 sites. This implies that not 4EBP1 but instead FGFR1 is the genetic target of Chr. 8p11 gains in breast and lung cancer. Accordingly, these tumors show increased sensitivity to FGFR1 and PI3K inhibition and this is a therapeutic vulnerability through restoring the tumor-suppressive function of 4EBP1. This research content is interesting, but the following changes that could be addressed to improve the manuscript.
1. The preface is not sufficient, and a related signal path diagram is welcome.

  1. It is necessary to improve the resolution of Figure 1 and Figure 3.
  2. When selecting inhibitors for research, BGJ398 is a pan-FGFR inhibitor, and ponatinib is a multi-kinase target inhibitor. Whether it is suitable for study FGFR1 using them. This requires the author to analyze whether it may be the result of the off-target effect of inhibitors.

Reviewer 2 Report

Authors proposed a paper entitled “Frequent 4EBP1 amplification induces synthetic dependence on FGFR signaling in cancer” for the publication in Cancer, mdpi.

This work is characterized by 15 pages, 5 composite figures, no tables and 36 references.

There are plenty of abbreviations and acronyms; therefore, I suggest adding an abbreviation list, according to the guidelines of this journal.

The paper deserves to be published; I only have some minor issues, listed in the following:

“All these studies suggest that 4E-BP1 act as a tumor suppressor by blocking oncogenic eIF4E and eIF4F activity. Surprisingly, Chr. 8p11-12 amplification coamplifies 4EBP1 along with FGFR1 suggesting that FGFR1 dependent phosphorylation of 4EBP1 might create a vulnerability to FGFR1 inhibitors in these tumors” this section requires the support of references cited from the relative literature.

In my opinion, the introduction deserves to be improved and expanded since the topic treated has a large scientific impact.

“In this study, we explored the specific effects of 4EBP1 on translation and therapeutic vulnerability generated by 4EBP1 amplification in breast and lung cancer.” This is the only sentence dedicated to the aims of this paper. In my opinion, this sentence could become an entire paragraph describing in details the aims of the authors with this paper.

“Cells were fixed” there is a double space before the starting of this sentence.

“4% paraformaldehyde in PBS for 15 min” is this on mass basis?

“0.25% Triton X-100 in PBS” see above.

“When tumours were between 80-100 mm3”. I would not use “were between”, but I would talk about “diameters/mean dimensions are …”. Moreover, mm3 should be with 3 at the apex.

“Data show results from at least…” double space before this sentence.

In the Results section, “4EBP1 is a target of genomic copy number gains (Chr. 8p11) in breast and lung cancer” seems to be a sub-title, but it is too long, in my opinion.

I suggest transforming Supplementary Fig. 1A in Figure 1, since the paper is characterized by 5 composite images.

4EBP1 amplified tumors show increased sensitivity to FGFR1 and PI3K inhibition” this is the second sub-title of the Results section. Please, check in the guidelines if these sub-titles should be numbered or whether they have a limit of words.

Try to improve the quality/focus of figure 3a.

“We performed de novo motif search using MEME suite”; my personal advice is to generally avoid personal expressions such as “we…”.

At page 12, authors write Figure 5. Maybe this is not necessary, since it is already reported in the caption. However, check the guidelines for this paper.

“and this need to be explored” should be “this needs to be explored”.

Corresponding author introduced only 3 self-citations (8% of total); therefore, this is surely acceptable.  
